



# Cloud droplet number enhancement from co-condensing NH₃, HNO₃, and organic vapours: sensitivity study

Yu Wang[1,2*], Beiping Luo[1], Judith Kleinheins[1], Gang I. Chen[3], Liine Heikkinen[4,5], and Claudia Marcolli[1*]

[1]Institute for Atmospheric and Climate Science, ETH Zurich, 8092, Zurich, Switzerland
[2]School of Geosciences, The University of Edinburgh, Edinburgh, UK
5   [3]MRC Centre for Environment and Health, Environmental Research Group, Imperial College London, UK
[4]Department of Environmental Science (ACES), Stockholm University, Stockholm, Sweden
[5]Bolin Centre for Climate Research, Stockholm University, Stockholm, Sweden

*Correspondence to*: Yu Wang (y.w@ed.ac.uk) and Claudia Marcolli (claudia.marcolli@env.ethz.ch)

10   **Abstract**

Semi-volatile compounds such as organics, nitrate, chloride, and ammonium are ubiquitous in atmospheric aerosols. Their gaseous precursors (organics, HNO₃, HCl, NH₃) co-condense with water vapour when ambient relative humidity (RH) increases, thus enhancing hygroscopic growth under sub-saturated conditions and facilitating activation as cloud condensation nuclei (CCN) to cloud droplets. In this study, we investigate the co-condensation effect on CCN activation for inorganics, organics, and their combination in a boreal forest site in autumn with our cloud parcel model that includes non-ideality of organic-inorganic mixtures. The volatility distribution of organics is highly uncertain but critically important to estimate the co-condensation effect. We compare two distinct volatility basis sets (VBS) established from experimental and modelling data at 25°C, which we amended with a volatility bin of saturation concentration $C^* = 10^4$ µg m⁻³, which proved to be highly relevant for CCN activation. The combined co-condensation of organics and inorganics increases CDNC by up to 52% in simulations initialized with RH of 80%, depending on VBS and updraft velocity during the air parcel uplifts. Non-ideality of the system is important for considering the co-condensation effect realistically. For the ideal case, the maximum CDNC enhancement due to the combined co-condensation effect is 131% while it is 52% for the non-ideal case. The combined enhancement in CDNC of inorganic and organic species exceeds the sum of individual effects and should be further constrained in different environments in cloud parcel models as a basis for regional and global simulations.



# 1 Introduction

Warm clouds play a key role in cooling Earth's climate by significantly reflecting shortwave radiation to space (L'Ecuyer et al., 2019). It has been a longstanding question of how aerosol particles influence the formation of clouds, their radiative effect, lifetime, and precipitation, exerting large uncertainty in climate assessment of the anthropogenic radiative effect forcing in recent Intergovernmental Panel on Climate Change (IPCC) reports (Forster et al., 2021; Myhre et al., 2013).

Aerosol particles can act as cloud condensation nuclei (CCN) and activate to cloud droplets through water condensation and form warm clouds when the air parcels are lifted in the atmosphere. Traditionally, Köhler theory (1936) is used to predict the CCN activation of aerosol particles (Köhler, 1936). That is, as the saturation ratio of water vapour ($S_w$) increases with adiabatic cooling, aerosol particles consisting of or containing non-volatile hygroscopic constituents take up water and grow in size to remain in equilibrium with the environment. When $S_w$ reaches the critical supersaturation ratio, which depends on aerosol size and chemical composition, aerosol particles activate and grow spontaneously to cloud droplets until the surrounding water vapour is sufficiently depleted (Köhler, 1936). However, not only water vapour, but also other condensable vapours (e.g. $NH_3$, $HNO_3$, HCl, organic compounds) can co-condense on aerosol particles together with water vapour hence enhancing hygroscopic growth and influencing cloud formation, referred to as "co-condensation effect".

The co-condensation effect may significantly influence local climate as semi-volatile compounds are ubiquitous in the atmosphere, but this effect is poorly constrained currently. It is a big challenge to assess hygroscopic growth and CCN activation while accounting for co-condensing semi-volatile compounds, because of unidentified evaporation losses of semi-volatile compounds during drying or heating processes in traditional instruments (Hu et al., 2018), such as HTDMAs (e.g., Duplissy et al., (2009) and CCN counters (e.g., Roberts and Nenes (2005)). Only a few studies investigated co-condensation of nitric acid (Rudolf et al., 2001) and simple organic surrogates (Rudolf et al., 1991; Hu et al., 2018) using modified instruments. Wang and Chen (2019) used an optics-based method and retrieved the hygroscopic growth factor of aerosol particles including co-condensation in ambient conditions in Delhi (India). This method was further combined with a thermodynamic model and revealed that co-condensation of HCl in Delhi contributed by 50% to the visibility reduction during haze events in winter and can halve the critical supersaturation for CCN activation (Gunthe et al., 2021). A similar co-condensation effect was found in Beijing but with $HNO_3$, which forms particulate nitrate in aerosol particles (Wang et al., 2020). Makkonen et al. (2012) included a nitric acid co-condensation scheme developed by Romakkaniemi et al. (2005) in a global climate model and found that including nitric acid co-condensation can increase the CDNC by 7% globally. It is challenging to assess the co-condensation effect of organics as these comprise more than tens of thousands of species and the total amount of ambient organic species in the gas and particulate phase cannot be fully detected.

To simulate co-condensation, models based on Raoult's law combined with the ideal gas equation have been developed in mass (e.g., Donahue et al. (2006)) and molar concentrations (e.g., Pankow (1994), Barley et al. (2009)) with complexity ranging from a single inorganic co-condensing species ($HNO_3$) (Kulmala et al., 1993) to considering 2727 organic compounds



(Topping and McFiggans, 2012). To account for organic compounds with volatilities differing by orders of magnitudes and aiming for simplicity in computation, Topping et al. (2013) assigned aerosols from different sources to 10 logarithmically spaced volatility bins and found a considerable enhancement of CDNC due to co-condensation in a cloud parcel simulation with different aerosol size distributions, organic aerosol types, and updraft velocities. The volatility basis sets were based on the thermodenuder measurement results by Cappa and Jimenez (2010), which include bins up to log(C*) of 3 and cover only

partially intermediate-volatility organic compounds (IVOC) and volatile organic compounds (VOC). Heikkinen et al. (2024) further extended volatility distributions up to log(C*) of 7 in their cloud parcel model by constructing them from the new FIGAERO-I-CIMS (Filter Inlet for Gases and AEROsols coupled to an iodide-adduct chemical ionization mass spectrometer) technique (Lopez-Hilfiker et al., 2014) to compute co-condensation of organic compounds in Hyytiälä, Finland. They found that IVOC/VOC (log(C*) >3) could significantly enhance the co-condensation effect, and that the aerosol size distribution has

a large influence on the magnitude of the effect with a nascent ultrafine mode enhancing it. However, the mass spectral peaks from FIGAERO-I-CIMS need to be calibrated with standard species and interpreted in terms of functional groups for an assignment to volatility bins, which could result in biases in volatility distributions (Heikkinen et al., 2024; Voliotis et al., 2022; Peng et al., 2023; Gkatzelis et al., 2021).

The studies performed by Topping et al. (2013) and Heikkinen et al. (2024) both based their conclusions on the assumption of

ideal mixing of the constituents in the condensed phase, which provides an upper estimate of the co-condensation effect and may be realistic in the case of an aerosol consisting of highly oxidised organic species. For a more reliable estimate of a mixed organic-inorganic aerosol, volatility and hygroscopicity information need to be combined. However, to the best of the authors' knowledge, no single study provides information on the volatility distribution of substances over a large vapour pressure range together with hygroscopicity information as required to simulate gas-particle partitioning taking solution non-ideality into

account.

Here, we integrated information from different experimental and modelling studies to set up two different VBS and investigate VBS sensitivity on co-condensation of organics. To simulate the co-condensation effect, we introduced new features into our model. Firstly, we recalculated the condensed mass of organics by accounting for the evaporation loss of semi-volatile compounds during sampling to estimate the mass loss and to provide a more accurate total organic mass for model initialisation.

Secondly, we developed our model to include solution non-ideality of organic-inorganic aqueous mixtures. With these new developments, we investigate the sensitivity of the co-condensation effect of organics and inorganics to volatility distribution, updraft velocity and non-ideality for a boreal forest site using field data from Hyytiälä, Finland.





## 2 Methodology

### 2.1 Input data for model initialisation

#### 2.1.1 Measured input

The data used to initialise the cloud parcel model was collected at the SMEAR II station in Hyytiälä, Finland, which is an atmospheric measurement supersite in a boreal forest (61°51' N, 24°17' E) (Hari et al., 2013; Heikkinen et al., 2024). The atmospheric composition of $PM_1$ was measured by a Quadrupole Aerosol Chemical Speciation Monitor (Q-ACSM) (Ng et al., 2011) as part of the Chemical On-Line cOmpoSition and Source Apportionment of fine aerosoL (COLOSSAL) project

(https://www.costcolossal.eu/) (Heikkinen et al., 2020; Chen et al., 2022). The Q-ACSM measures mass concentrations of non-refractory $PM_1$ species, including ammonium ($NH_4^+$), nitrate ($NO_3^-$), chloride ($Cl^-$), sulfate ($HSO_4^{2-}$, $SO_4^{2-}$), and organics (OA). Chen et al. (2022) resolved the measured OA in Hyytiälä into two components using Positive Matrix Factorisation (PMF) combined with the multilinear engine (ME-2) algorithm: less-oxidized oxygenated organic aerosol (LO-OOA) and more-oxidized oxygenated organic aerosol (MO-OOA). In this study, we focus on the autumn (September-November) of 2018.

The seasonally averaged chemical composition for initialising the cloud parcel model is shown in Table 1. Equivalent black carbon (BC) during the COLOSSAL campaign was measured using the Aethalometer AE33 (Drinovec et al., 2015) at 880 nm wavelength in the default configuration and the data were post-processed as suggested by the manufacturer (see Chen et al., 2022).

The co-located aerosol size distribution and meteorological parameters during the COLOSSAL campaign were collected from

the online database of the SMEAR II station (https://smear.avaa.csc.fi/download, last access: 15 March 2024). The campaign average of the aerosol size distribution (diameter range: 2.82–1000 nm) measured by a Differential Mobility Particle Sizer (DMPS; Aalto et al., 2001) can be seen in Fig. 1. The mean ambient temperature and relative humidity (RH) measured at an altitude of 16.8 m by the meteorological platform during the campaign were 8 °C and 80%, respectively, the surface pressure was 991 hPa. As there were no measurements of $NH_3$ and $HNO_3$ performed in autumn 2018, we used concentrations measured

in Autumn 2010 (Makkonen, 2014). The mass concentration of gaseous organic compounds is calculated by the cloud parcel model, as detailed in Sect. 2.3, based on the VBS of total organics described in Sect. 2.1.2. To account for non-ideality of the organic molecules, we used AIOMFAC to simulate the activity coefficients of representative organics in each volatility bin (for details see Sect. 2.2).

Note that the Q-ACSM and DMPS instruments were stationed indoors at room temperature while the average ambient

temperature was 8 °C. Assuming an indoor temperature of 25 °C, the 17 °C temperature increase when the air is sampled into the instruments causes evaporation of semi-volatile compounds. In addition, drying of the aerosol before entering the measurement devices further contributes to evaporation loss. We modelled this loss with the cloud parcel model (details in Sect. 2.4.1) and accounted for it when calculating back the total organic mass used in model initialisation.





**Table 1**. Atmospheric conditions and concentrations used in this study: Mass concentrations of inorganics and organic components (condensed phase, including LO-OOA and MO-OOA) and BC; meteorological parameters; and gas phase concentrations of inorganic compounds.

| Condensed-phase compound concentrations [$\mu g\ m^{-3}$] | |
|---|---|
| $NH_4^+$ | 0.19 |
| $NO_3^-$ | 0.19 |
| $Cl^-$ | 0.01 |
| $HSO_4^-$ , $SO_4^{2-}$ | 0.41 |
| LO-OOA | 0.48 |
| MO-OOA | 1.28 |
| BC | 0.17 |
| **Meteorological parameters** [mean $\pm$ std] | |
| RH at 16.8 m [%] | $80 \pm 31$ |
| T at 16.8 m [°C] | $8 \pm 6$ |
| Surface pressure [hPa] | $991 \pm 11$ |
| **Gas-phase inorganic compound concentrations [$\mu g\ m^{-3}$ (ppb)]** | |
| $NH_3$ | $0.10 \pm 0.06\ (0.14 \pm 0.09)$ |
| $HNO_3$ | $0.16 \pm 0.08\ (0.06 \pm 0.03)$ |





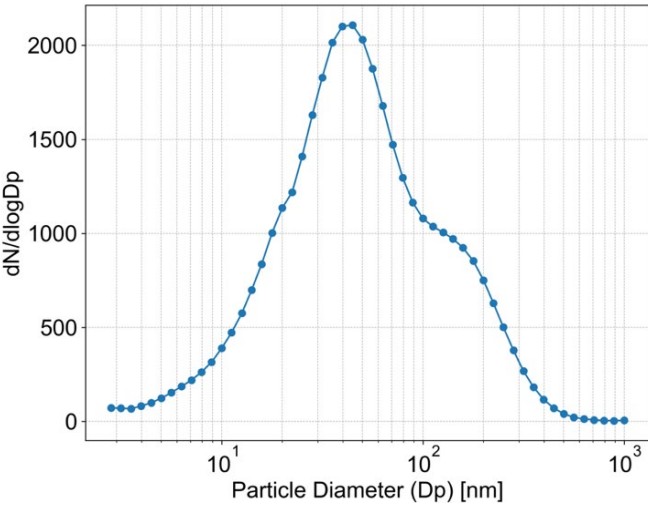

**Figure 1**. Campaign-averaged particle size distribution during September-November 2018. The total number of particles with diameters from 2.82 nm to 1000 nm is 1871 cm$^{-3}$.

### 2.1.2 Aerosol volatility distribution

To set up two generic VBS for MO-OOA and LO-OOA, we consider experimental and modelling studies (Lane et al., 2008; Tsimpidi et al., 2010; Koo et al., 2014; Stirnweis et al., 2017) in combination with information about hygroscopicity of these OA components (Ciarelli et al., 2017) (see Sects. S1 and S2 for more information). The two VBS used in this study follow the trend found in experimental and modelling data (Table S1) of increasing mass per bin with increasing volatility. In our distributions, the bin with saturation concentration $\log(C^*) \leq -1$ is the least volatile one and sums up compounds with $C^* \leq 0.1$ µg m$^{-3}$, which can be considered as fully partitioned to the particulate phase at $T \leq 25$ °C. Published VBS , traditionally derived from aerosol mass spectrometer data (Tsimpidi et al., 2010; Cappa and Jimenez et al., 2010; Stirnweis et al., 2017) include bins up to $\log(C^*) = 3$ and thus comprise compounds with relevant partitioning to the particulate phase at 25 °C, but they lack substances that start to condense when temperature is lower. To account for the low temperatures prevailing at Hyyitälä in autumn, we therefore extended the VBS to $\log(C^*) = 4$ by extrapolating the mass per bin for $\log(C^*) = 0–3$ to $\log(C^*) = 4$. Figure 2 shows the two normalized VBS that we derived. Hereafter, simulations with both VBSs are performed, and impact of co-condensation is compared to assess the role that the assumptions concerning the volatility distribution play in co-condensation. The absolute values (mass concentrations) were determined during preprocessing considering evaporative loss in the inlet tubing (Sect. 2.4).

For explicit modelling of representative substances for LO-OOA and MO-OOA, chemical information in addition to volatility is required. We orient ourselves here on O:C ratios reported for LO-OOA and MO-OOA. Yet, reported values for these OA



components vary strongly between studies, namely from 0.48 to 1.08 for MO-OOA and from 0.27 to 0.66 for LO-OOA (Setyan

et al., 2012; Sun et al., 2012; Xu et al., 2018; Zhu et al., 2021). Only very few studies report volatility bin resolved O:C ratios

(Koo et al., 2014; Ciarelli et al., 2017). Here we use the O:C and molecular weight information provided by Ciarelli et al.

(2017), and identify their SOA set 2 with LO-OOA and their SOA set 3 with MO-OOA. In the next step, we defined model

compounds to represent the thus constrained volatility bins (see Tables S2 and S3).

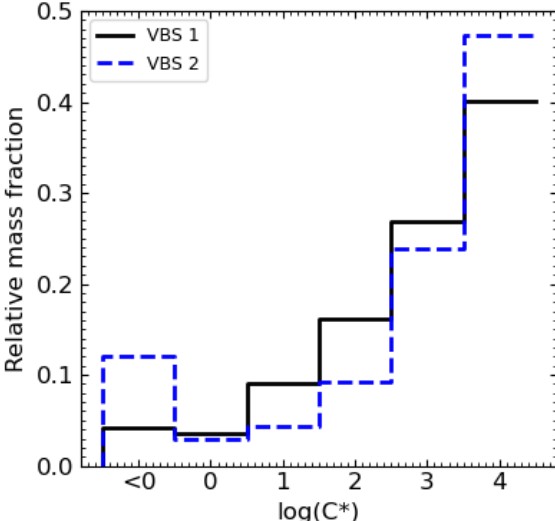


**Figure 2**: The two generic volatility basis sets VBS 1 and VBS 2 derived at 25°C and used in this study to represent the total

OA (gas and condensed phase).

**2.2 Physico-chemical properties of the organic fraction: Pre-calculations with AIOMFAC**

Solution non-ideality of internally mixed organic-inorganic particles has been found to influence their phase state, their

hygroscopic growth and the gas-particle partitioning of semi-volatile organic compounds (Zuend and Seinfeld, 2012). To

accurately estimate co-condensation of $NH_3$, $HNO_3$, $HCl$, and organic vapours (LO-OOA and MO-OOA) with the parcel

model, activity coefficients of all compounds and as a function of RH are required. With the group contribution model

AIOMFAC, gas–particle partitioning in equilibrium can be calculated considering non-ideality of organic–inorganic mixtures

(Zuend et al., 2008; Zuend et al., 2011; Zuend and Seinfeld, 2012; Zuend and Seinfeld, 2013). Integrating AIOMFAC within

the parcel model would be computationally expensive, therefore AIOMFAC was used in a pre-processing step to create a look-

up table for activity coefficients of water and the organic fractions as a function of RH. Activity coefficients of the inorganic

compounds were directly calculated in the parcel model with the Pitzer model as described in Sect. 2.3 below.



To set up the look-up table, 2–3 compounds were selected per volatility bin leading to two sets of 15 model compounds to represent LO-OOA and MO-OOA, respectively, as described in Sect. S2 in the Supplement. For these model compounds
AIOMFAC calculations as a function of RH were performed and the average molecular weight in each volatility bin of LO-OOA and MO-OOA was determined, which is another input parameter required for the parcel model. To obtain the total mass (gas + particle phase) in each bin, the total mass concentrations of the organic fractions together with the inorganic salts with concentrations as listed in Table S2 were equilibrated between the gas and the condensed phase at dry conditions (RH = 20 %) and adjusted to match the condensed organic mass concentrations measured by the Q-ACSM in Hyytiälä (see Table 1). The
so-derived organic-inorganic model system was then equilibrated at RH from 20 % to 99.9999 % to determine the activity coefficients of water and the organic model compounds at each humidity step. To derive representative activity coefficients per bin, the activity coefficients of the individual substances in each bin were averaged. These activity coefficients as a function of RH are shown in Fig. 3 and further details are given in Sect. S4 in the Supplement. Note that the higher activity coefficients calculated for LO-OOA reflect the lower hydrophilicity and polarity of these substances compared with the more hydrophilic
MO-OOA.

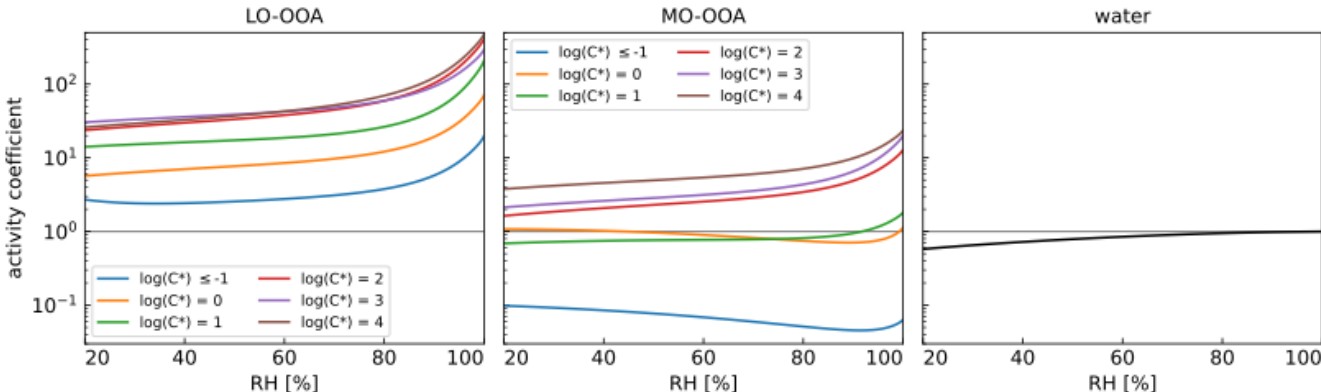

**Figure 3**. Average activity coefficients of the six volatility bins for LO-OOA (left panel) and MO-OOA (center panel) and of water (right panel) as a function of RH for the mixed organic-inorganic model particles calculated with AIOMFAC in the pre-processing. The horizontal grey line at $10^0$ corresponds to solution ideality.

**2.3 The adiabatic cloud parcel model**

We employed the comprehensive microphysical box model ZOMM (Zurich Optical and Microphysical box Model) developed at ETH Zurich (Luo et al., 2003a; Luo et al., 2003b; Luo et al., 2003c; Brabec et al., 2012). ZOMM is a cloud parcel model that is able to dynamically simulate adiabatic ascents of air parcels and the formation of cloud droplets and ice crystals. The model is Lagrangian in radius space, i.e., particle sizes are allowed to evolve freely in radius space, thus avoiding numerical
problems, and it can be initialised with any particle size distribution. It is able to account for reduced diffusion in condensed phase by using radial shells of variable thickness, and it is designed with variable time steps during the ascent and flexible bin width for the aerosol size distribution to minimise computational time. As we assume fast liquid-phase diffusion for RH ≥ 80%



leading to particles with homogeneous composition, we do not require radial shells setup here. To describe solution non-ideality, the activity coefficients of $H^+$, $NH_4^+$, $Cl^-$, $NO_3^-$, $SO_4^{2-}$, and $HSO_4^-$ ions are calculated using the Pitzer ion-interaction

model (Carslaw et al., 1995; Luo et al., 1995; Clegg et al., 1998). The ones of the organic volatility bins LO-OOA and MO-OOA are taken from lookup tables generated with AIOMFAC (see Sects. 2.2 and S4).

During the ascent, adiabatic cooling drives water vapour and the gas-phase species ($NH_3$, $HNO_3$, HCl, organics) to partition into the condensed phase. The flux of the inorganic species $i$ and of the organic compounds in volatility bin $i$ from the gas phase to the particle surface ($j_{surf,i}$) is calculated using the gas-phase diffusion onto a spherical particle with radius $r_g$ in steady

state as (Pruppacher and Klett, 2010):

$$j_{surf,i} = 4\pi r_g D_{g,i} \frac{p_i^{vap}(T_p) - p_i}{RT_{air}} \tag{1}$$

with $D_{g,i}$ being the gas-phase diffusion coefficient of species $i$, and $p_i^{vap}(T_p)$ and $p_i$ the vapour pressure above the particle and the gas-phase partial pressure of species $i$, respectively. Note that the model accounts for condensational heat release by discriminating between the particle temperature ($T_p$) and the ambient temperature ($T_{air}$). The particle temperature is

determined by the latent heat release of water vapour condensation and the heat conduction of air (Pruppacher and Klett, 1997).

The vapour pressure above the particle $p_i^{vap}(T_p)$ of the organics in volatility bin $i$ are calculated as:

$$p_i^{vap}(T_p) = p_{i,0}(T_p) \, x_i \, \gamma_i \, K_i \tag{2}$$

Here $x_i, \gamma_i$, and $K_i$ are the mole fraction, activity coefficient of species $i$, and the Kelvin effect, respectively. The pure

component vapour pressure of the organics in the volatility bin $i$, $p_{i,0}$ is derived from the nominal vapour concentration $10^i$ µg m$^{-3}$, of the volatility bin $i$ as

$$p_{i,0}(T_p)(in\ Pa) = 10^i \times 10^{-6} \times \frac{RT_{air}}{M} \times \exp\left(-\frac{H_{vap}}{R}\left(\frac{1}{T_p} - \frac{1}{298.15}\right)\right) \tag{3}$$

Where $M$ is the molar mass in g mol$^{-1}$ listed in Table S4 and $H_{vap}$ is the heat of vaporisation (given in Table S5) using Clausius-Clapeyron to calculate the temperature dependence. The activity coefficients have been pre-calculated with AIOMFAC as

described in Sects 2.2 and the lookup table (Table S7) in S4.

The Kelvin effect $K$ is given by



$$K_i = \exp\left(\frac{2\sigma V_i}{RT_p}\right) \tag{4}$$

$\sigma$ is the surface tension, taken here as a constant value of 72 mN m$^{-1}$. $V_i$ is the molar volume of species $i$.

The water vapour pressure $p_w$ over the bulk solution is given by $p_{w,0} \times a_w$, where $p_{w,0}$ is the H$_2$O vapour pressure over pure water (Murphy and Koop, 2005). The water activity is obtained as the product of the contribution of ions and organic species:

$$a_w = a_{w,inorg} \times a_{w,org} \tag{5}$$

$$a_{w,org} = x_{w,org}\, \gamma_{w,org} \tag{6}$$

$x_{w,org}$ is the mole fraction of water calculated with organic species only. $\gamma_{w,org}$ is the activity coefficient of water taken from the lookup Table S7.

The $a_{w,inorg}$ of H$^+$, NH$_4^+$, Cl$^-$, NO$_3^-$, SO$_4^{2-}$ . and HSO$_4^-$ are calculated using the Pitzer ion-interaction model (Carslaw et al., 1995; Clegg et al., 1998; Kienast-Sjögren et al., 2015; Luo et al., 2023). The vapour pressures and dissociation into ions of H$_2$O, NH$_3$, HNO$_3$, and HCl are calculated using the Henry's law coefficients and equilibrium constants as given in Luo et al. (2023), together with the activity coefficients calculated with the Pitzer-ion interaction model, and taking the Kelvin effect into account.

We assume that the campaign-averaged aerosol size distribution is an internal mixture of ammonium bisulfate, ammonium sulfate, ammonium nitrate, ammonium chloride, LO-OOA, MO-OOA, and black carbon, as given in Sect. 2.1. Total mixing ratios for NH$_3$, HNO$_3$, and HCl for the parcel model are calculated as the sum of measured gas and particle concentrations. Total mass concentration of LO-OOA and MO-OOA are calculated back from the measured condensed mass by Q-ACSM as explained in Sect. 2.4. The binned microphysics scheme for the size distribution is used with a moving centre comprising variable size bins with widths between 10 nm to 0.5 μm in dry radii, described by the geometric mean radii and number concentration of all species in each bin. The parcel starts at ground level. A series of cooling rates and corresponding updraft velocities are simulated, ranging from 0.27 K h$^{-1}$ (0.0075 m s$^{-1}$) to 180 K h$^{-1}$ (5 m s$^{-1}$), which covers the typical updrafts of stratus to stratocumulus clouds.

## 2.4 Composition initialisation

To simulate cloud droplet activation, we initialise the model at ground level with an ambient temperature of 8°C, relative humidity of 80 %, pressure of 990.8 hPa, and the size distribution given in Fig. 1. We take the aerosol composition (condensed and gas phase) as the one shown in Table 1. To comply with the volatility distribution of organics according to the generic



VBS 1 and 2, and to take the mass loss due to heating and drying into account, the LO-OOA and MO-OOA mass concentrations measured by the Q-ACSM were back-calculated to the total organic mass in the gas and in the condensed phase (see Sect. 2.4.1). To match the total mass concentration according to the size distribution, the aerosol composition was complemented with $Na^+$, mineral dust and black carbons as described in Sect. 2.4.2.

### 2.4.1 Evaporation due to temperature change and drying processes during sampling

To exemplify expectable condensed mass losses when sampling occurs at low ambient temperatures and to initialize the cloud parcel model, we estimate the evaporation loss for the temperature change from the autumn mean ambient 8°C to indoors 25°C and the drying before entering the instrument's inlet by simulating the heating and drying of the aerosol flow on its way to the instrument's inlet. To this end, we estimate the residence time in the sampling line before the dryer as ~7 s by assuming a length of 3.5 m, ¼ inch inner diameter, and 1 L/min flow representing a typical value. Note that in Hyytiälä a 3 L/min overflow was used to minimize losses in the sampling line (Heikkinen et al., 2020). Considering the two commonly used drier types: silicon dryers (½ inch, 60 cm) and Nafion dryers (0.7 inch, 60 cm), we take the average of these dryers ( ~7 s) as the overall residence time. We use the cloud parcel model to simulate the evaporation of MO-OOA and LO-OOA during the sampling process and to quantify the evaporation losses, such that the calculated concentrations at the Q-ACSM inlet correspond with the values measured by the instrument. Figure 4 shows that the heating leads to a strong decrease of RH to about 37 % followed by a further decrease to a value of 20 %, which we assume to be the RH within the dryer. With these assumptions, the evaporation loss before the dryer due to heating is ~10 % followed by an additional ~5 % in the dryer adding up to a total of ~15 % for both, MO-OOA and LO-OOA before the Q-ACSM is reached. In the cloud parcel simulation for VBS 1, we use the total ambient organic mass (gas + particle phase, MO-OOA = 7.28 μg m$^{-3}$ and LO-OOA = 6.17 μg m$^{-3}$) at time 0 in Fig. 4 and the particle mass MO-OOA = 1.52 μg m$^{-3}$ and LO-OOA = 0.60 μg m$^{-3}$. The corresponding values for VBS 2 are as follows: total MO-OOA = 6.42 μg m$^{-3}$, LO-OOA = 3.16 μg m$^{-3}$ and ambient particular mass MO-OOA = 1.412 μg m$^{-3}$ and LO-OOA = 0.514 μg m$^{-3}$. The concentrations of ambient particle organics are higher than the measured Q-ACSM data, as they evaporate partly in the inlet and dryer as shown in Fig.4.



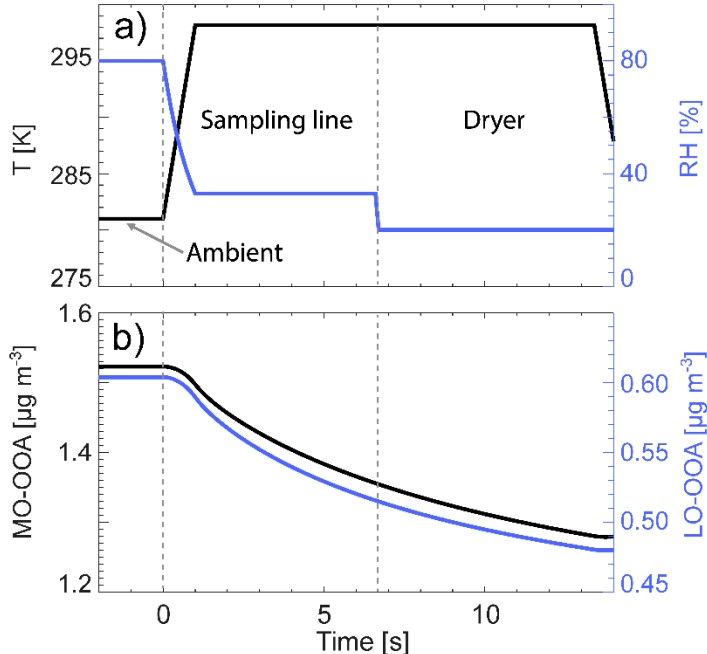

**Figure 4.** Time series of a) RH and temperature and b) the condensed mass concentration of MO-OOA and LO-OOA in the sampling line and the dryer. The mass concentration at 14 s corresponds to the measured, and the one at 0 s to the estimated ambient mass concentration.

### 2.4.2 Adjustment of total mass concentration to the measured size distribution

We take the concentrations of inorganic species both in the condensed phase ($HSO_4^-$, $SO_4^{2-}$, $NO_3^-$, $Cl^-$, $NH_4^+$) and gas phase ($NH_3$, $HNO_3$) as the one given in Table 1. In order to represent the sea salt aerosol contribution, which cannot be measured by Q-ACSM, we assume a sodium/sulfate ratio of 0.2. With this amendment, the total volume of condensed organic (LO-OOA and MO-OOA) and inorganic species is 86.4 % of the aerosol volume shown in Fig.1. To achieve consistency, we assume the missing volume to be mineral dust (10 % v/v) and black carbon (3.6 % v/v), with the mineral dust forming the tail of the particle size distribution with radius > 250 nm and possessing a 10 nm liquid coating with the same composition as the liquid particle composition, and the black carbon being internally mixed in each size bin.

Like this, the initial aerosol composition used for the parcel model (shown in Table S8) matches the size distribution of Fig. 1. The aerosol is equilibrated for 30 min at ambient conditions (80% RH) with the total gas phase given in Table 1 (inorganic species) and estimated in Sect. 2.4.1 (organic species). In order to save computational time, we use an updraft of at least 1.2 m s$^{-1}$ for RH < 98% when the simulated updrafts are smaller (e.g., for an updraft of 0.3 m s$^{-1}$, the updraft of 1.2 m s$^{-1}$ was used for the first 300 s up to 98% RH ), and then the simulation is continued with the set updrafts when RH > 98%. The initially





higher updraft has a negligible influence on the final CDNC as most of the uptake of $H_2O$ and other trace gases occurs at RH
> 98%.

## 2.5 Equivalent κ values

Petters and Kreidenweis (2007) proposed to use a single parameter κ to characterise the aerosol hygroscopicity for non-volatile
aerosol particles, which can be calculated from the growth factor ($GF$) as follows:

$$GF = \frac{r_{wet}}{r_{dry}} \tag{7}$$

$$\kappa = (GF^3 - 1)\left(\frac{1}{a_w} - 1\right) \tag{8}$$

Where $r_{dry}$, $r_{wet}$, and GF are the initial dry particle diameter, wet diameter (dry diameter plus absorbed water), and growth
factor at water activity $a_w$., respectively.

Here, to account for co-condensing semi-volatile materials, we used an equivalent κ, which includes the contribution of co-
condensing vapours to $GF$. In our calculation of equivalent κ values from the model output, $r_{wet}$ is the sum of the initial dry
diameter, absorbed water, and co-condensed semi-volatile materials.

## 3 Results

The cloud droplet formation enhancement induced by co-condensing semi-volatile compounds is sensitive to the
environmental conditions. This section shows the sensitivity of the co-condensation effect to the VBS of organic compounds
(Sec. 3.1), updraft velocity (Sec. 3.2), and non-ideality of organic compounds (Sec. 3.3).

### 3.1 Volatility basis set of organic compounds

In Fig. 5, the two different VBS for the total organics are compared in cloud parcel model simulations (for detailed information
on the VBS see Sec. 2.1.2). The black lines in Fig. 5 are the volatility distribution of total organic mass for VBS 1 and VBS 2.
The coloured lines show the modelled organic mass in the condensed phase at different RH for an updraft velocity of 0.3 m s⁻
¹. For both VBS, the organic mass in the bins with log(C*) ≤ -1 and 0 partition completely to the condensed phase at the lowest
displayed RH of 80 %, while the species present in the bin with log(C*) = 4 only partition substantially to the condensed phase
when the maximum RH during droplet activation is reached. Note that the calculated total organic mass for VBS 1 is higher
by 40 % (see legend of Fig. 5) than that of VBS 2 because of its lower share of mass present in the bin with log(C*) ≤ -1,
which results in a higher back-calculated gas-phase organic mass for the same amount of measured condensed-phase organic



mass. Therefore, the condensed-phase organic mass in VBS 1 at all RH levels is higher than that in VBS 2, approximately 4

% at 80 % RH to 44–47 % above 99.5 % RH. Thus, for VBS 1, a higher CDNC enhancement by co-condensation is expected.

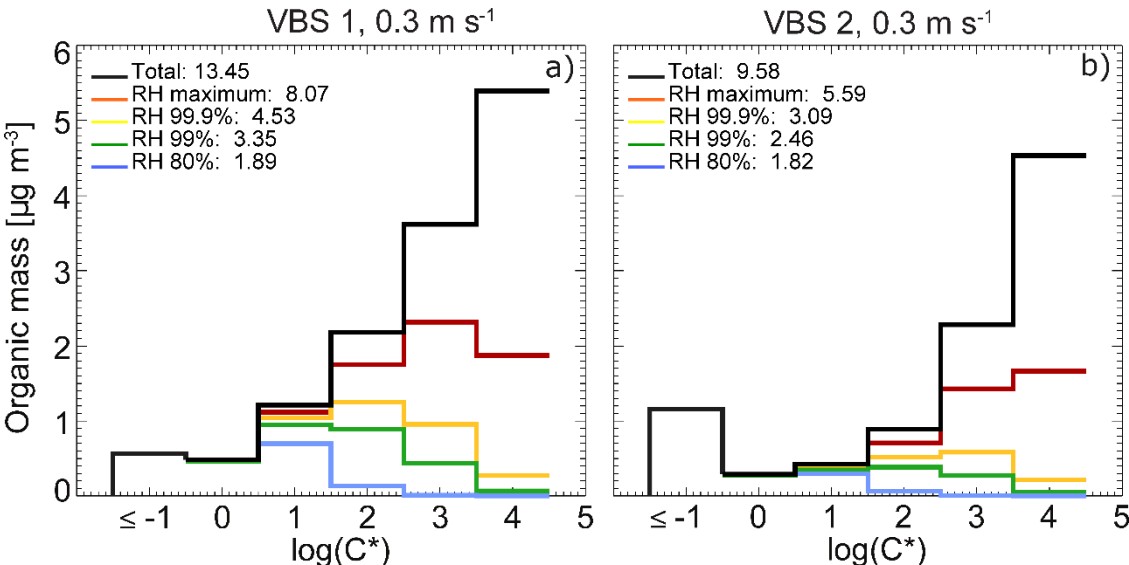

**Figure 5**. Total organic mass (gas and condensed phase: black line) and organic mass in the condensed phase at selected RH (coloured lines) summed up (masses given in the legend) and per volatility bins in the VBS derived in this study based on a) Tsimpidi et al. (2010) (VBS 1) and b) Cappa and Jimenez (2010) (VBS 2).


Figure 6 shows an overview of the time evolution of supersaturation, partitioning of semi-volatile compounds between gas and condensed phase, growth of aerosol particles and cloud droplets during the adiabatic cooling process for VBS 1 and VBS 2 at an updraft of 0.3 m s$^{-1}$. As shown in panels a) and b), the two VBS cases reach similar maximum supersaturations (~0.23%) and show a similar time evolution of the co-condensing inorganic vapours $NH_3$, $HNO_3$, and $HCl$ to form the particulate species

$NH_4^+$, $NO_3^-$, and $Cl^-$. Stronger differences arise for the total condensed mass of MO-OOA (panels c) and LO-OOA (panels d) due to the different volatility distributions. Due to the higher total organic mass (Fig. 5), VBS 1 has more organic mass available in the gas phase than VBS 2 (panels c and d), resulting in a higher condensed mass concentration just after cloud formation (see panels e). The increase in organic condensed mass with time is driven by the bins with higher volatilities, with the increase in condensed phase following the sequence of volatility. Due to the higher hygroscopicity of MO-OOA compared to LO-OOA,

for a given RH, MO-OOA compounds show a higher partitioning to the condensed phase than LO-OOA compounds for the same volatility bin. Therefore, organics with $\log(C^*) = 1$ of MO-OOA are fully in the condensed phase already at the starting conditions (RH = 80 %), while those of LO-OOA require a larger condensed mass to partition completely to the particulate phase. The organic compounds in the volatility bin of $\log(C^*) = 4$ (light green lines, labelled '4') show a strong co-condensation





effect, resulting in a large contribution to the condensed-phase organic mass at cloud droplet activation, although they were
thought too volatile to co-condense previously (Topping et al., 2013). Similar results were found in Heikkinen et al. (2024).

The co-condensation of organic and inorganic compounds increases the solute mass of the droplets, which can be expressed
as a relative increase in dry volume and dry radius of the droplets compared to the initial state at ground level (time = 0 s, RH
= 80%). This relative dry volume increase is shown in panels e) of Fig.6. Until activation, the dry volume increased by a factor
of 2.1 and 1.8 for VBS 1 and VBS 2, respectively (dashed black line). After activation of the droplets, semi-volatile compounds
keep partitioning into the droplets, such that at the end of the simulation, the droplet dry volume has even increased by a factor
of 3.5 and 3.1 for VBS 1 and VBS 2, respectively (solid black line), driven by the strongly increased condensed mass due to
water vapour condensation. Consistently, the ultimate dry radius increases by over 100 % during cloud formation for both
VBS cases, with a higher increase for VBS 1, as shown by the colour-coding in panels f).

In panels b–e), the vertical dashed line marks the time of maximum supersaturation in the air parcel. Around this time,
activation of aerosol particles occurs. The exact activation time for each size bin, as defined by its critical supersaturation, can
deviate slightly from this time, as shown in panels f) with black dots. Note that particles of intermediate size  activate first
because smaller particles require a higher supersaturation to activate, while larger particles require more time to reach their
activation size due to the limitation in gas-phase flux to the particulate surface. After activation, the droplets in the activated
size bins grow rapidly, thereby decreasing humidity in the air parcel. This hinders smaller size bins from activating, while all
larger particles are still able to activate because of their lower critical supersaturation. Some intermediate-sized bins that have
been activated at one point, even shrink again to aerosol particles. Due to this dynamic competition for water vapour, the
activated cloud droplet number concentration depends strongly on the updraft velocity of the air parcel, as discussed in the
next section.









**Figure 6.** Time series of a) supersaturation, b–d) inorganic and organic species concentrations, e) droplet dry volume relative to the initial state of RH = 80%, and f) wet radius, for VBS 1 (left panels, a1–f1), and for VBS 2 (right panels, a2–f2), at an updraft velocity of 0.3 m s$^{-1}$. The numbers in panels c) and d) refer to the log(C*) bins of the organic compounds. The volatility bin of log(C*) <=-1 is not shown as it is condensed phase only and therefore does not show RH dependent partitioning. Panel e) shows co-condensation (CC) up to activation (maximum supersaturation, dashed line) and continued to the end of simulation (solid line). The colour-coding in panel f) shows the increase in the dry radius of the particles relative to t = 0 s as given in the legend. The vertical dashed line marks maximum supersaturation of the air parcel as shown in panel a).

## 3.2 Updraft velocity

In this section, we investigate the influence of co-condensation on the CDNC at various updraft velocities (and cooling rates assuming adiabatic expansion of the air parcel) from 0.0075 m s$^{-1}$ (0.27 K h$^{-1}$) to 5 m s$^{-1}$ (180 K h$^{-1}$). Figure 7 analyses the CDNC with co-condensation of organics, inorganics, and the combination of both for the two VBS. Generally, as more aerosol particles activate to cloud droplets with increasing maximum supersaturation in higher updrafts, the CDNC increases with updraft velocity (Fig. 7a), resulting in a higher activated fraction (AF) of aerosol particles to cloud droplets (panels b) for all co-condensation scenarios in the two VBS cases. There is a clear enhancement in CDNC by 21-52% and 18–39% induced by the combined co-condensation effect of inorganics and organics for VBS 1 and VBS 2, respectively (panels c). Correspondingly, the AF increases by 1.1–15.9% and 1.0–12.4% due to the combined co-condensation effect for VBS 1 and VBS 2, respectively. Namely, the AF for the control case with no co-condensation (in black) ranges from 2.8% to 56.8%, depending on updraft velocity, and it increases to 3.9% to 68.6% due to combined co-condensation for VBS 1. For VBS 2, the AF ranges from 2.6% to 56.4% for the control case and increases to 3.6% to 66.3% for combined co-condensation. Herein, the co-condensation of inorganic compounds is comparable for the two VBS, as both cases use the same input for the inorganic species. For co-condensation of organic compounds, the larger condensable organic mass of VBS 1 results in a stronger enhancement of cloud droplet formation across all updrafts. So, the overall co-condensation effect in VBS 1 is larger than in VBS 2. It is worth noting that the increase in CDNC for combined inorganic and organic co-condensation is not just the sum of organic and inorganic condensation simulated separately, but exceeds it for intermediate updrafts. For example, the maximum CDNC enhancement of the overall co-condensation effect for VBS 1 is 52%, while 19% and 28% for the inorganic and organic co-condensation effects separately. These synergistic effects arise because condensed mass causes even more mass to condense.

Interestingly, the two VBS show similar non-linear updraft-dependent trends in CDNC changes in Fig. 7c: with increasing updraft velocity, the CDNC enhancement due to the combined co-condensation effect declines first, reaches a minimum at 0.03–0.05 m s$^{-1}$ followed by an increase and a maximum at around 0.3–0.5 m s$^{-1}$, then decreases again. The first declining trend and the minima are only present for the simulations including organic compounds, whereas the decrease in CDNC enhancement at high updrafts is present in all co-condensation simulations. The latter occurs because the potential for CDNC enhancement declines when particle activation approaches AF = 1. The CDNC enhancement at low and intermediate updrafts



may depend on different factors, including volatility, aerosol number size distribution, total condensed and condensable mass
and time available for co-condensation (Heikkinen et al. 2024). As the effects of these factors on CDNC enhancement are
nonlinear they result in a complex overall trend that we will further explore for three representative updraft velocities (0.03,
0.3, and 3 m s$^{-1}$), which represent the minimum, maximum, and the declining part at high updrafts of CDNC enhancement.

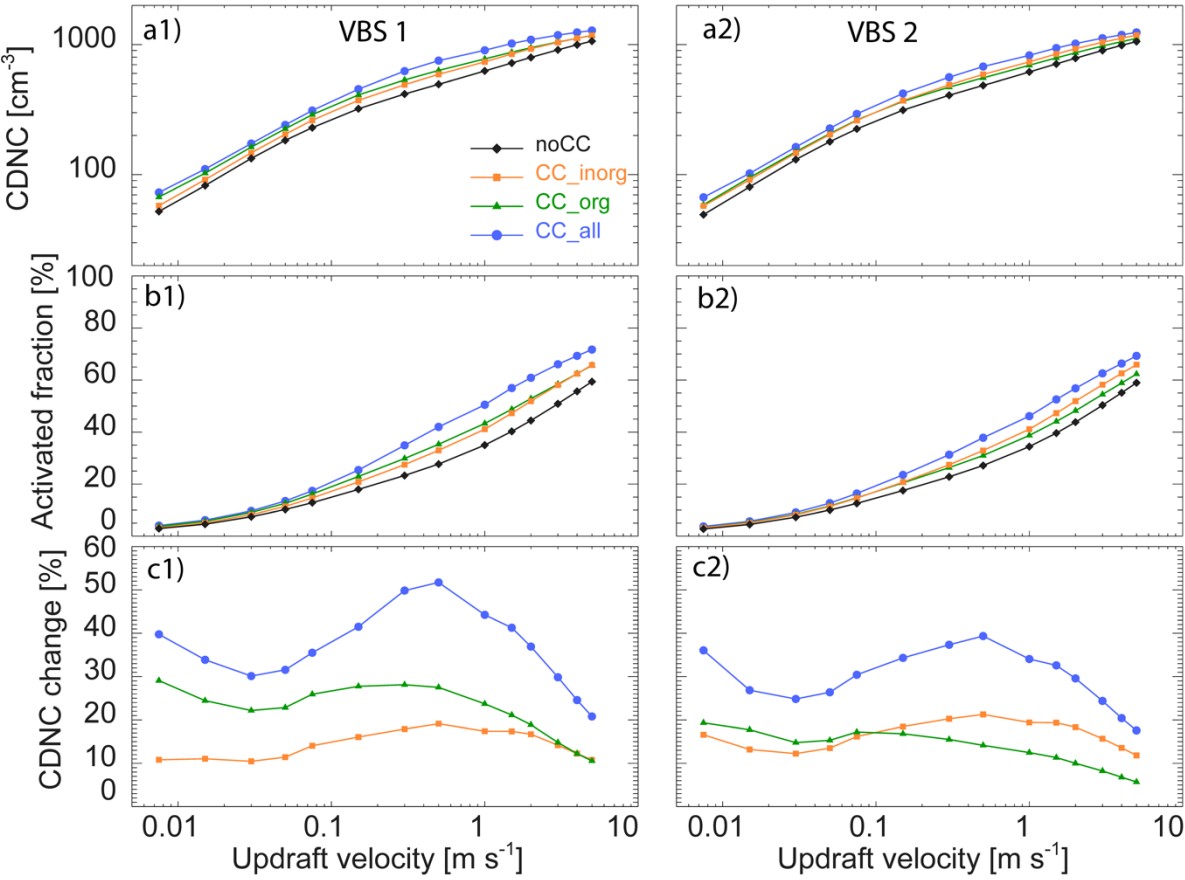

**Figure 7.** CDNC (panels a), activated fraction (panels b), and CDNC change relative to the case without co-condensation
(panels c) as a function of updraft velocities for different co-condensation scenarios for VBS 1 (left panels) and VBS 2 (right
panels).

Figure 8 shows the size-dependent dry radius ratios and equivalent $\kappa$ at the activation points (time when a particle has reached
its critical supersaturation) for the three selected updraft velocities. As the two VBS exhibit the same trend, we focus on VBS
1 in the following discussion.





The case with an updraft of 0.3 m s$^{-1}$ shows the strongest dry radius increase of up to a factor of 1.68 due to co-condensation, followed by the updrafts of 0.03, and 3 m s$^{-1}$ with increases up to 1.55 and 1.38, respectively (Fig. 8a). As a result, the maximum equivalent $\kappa$ values increase the most at the updraft of 0.3 m s$^{-1}$ from 0.16 for control to 0.84 for co-condensation, followed by 0.60, and 0.50 for the updrafts of 0.03 and 3 m s$^{-1}$, respectively (Fig. 8c), indicating that a higher increase in equivalent $\kappa$

correlates with a higher CDNC increase. In comparison, in the absence of co-condensation, $\kappa$ values remain almost constant. For the updraft of 0.03 m s$^{-1}$, the particles with the highest equivalent $\kappa$ enhancement are the ones with $r_{\mathrm{dry}}$(RH=80%) = 111 nm, for 0.3 m s$^{-1}$ the ones with $r_{\mathrm{dry}}$(RH=80%) = 41 nm, and for 3 m s$^{-1}$ the ones with $r_{\mathrm{dry}}$(RH=80%) = 21 nm. Interestingly, these radii almost coincide with the $r_{\mathrm{dry}}$ of the smallest particles that are still activating thanks to co-condensation (see line shadings in Fig. 8). In the absence of co-condensation, the demarcation between activating and non-activating particles is at

larger radii as indicated by the symbols in Fig. 8. The difference in radius between the smallest still activating particles with and without co-condensation is largest for the updraft of 0.3 m s$^{-1}$ (18 nm in radius), while it is 14 nm for 0.03 m s$^{-1}$, and only 7 nm for 3 m s$^{-1}$. This radius difference together with the number of particles in this size range as given by the particle size distribution determines the CDNC enhancement, confirming that it is not just a single parameter but a complex interplay of updraft velocity, condensed and condensable mass, and aerosol size distribution that determine the effect of co-condensation.

Note, that these parameters are the same as identified by Heikkinen et al. (2024) as being key for the magnitude of the co-condensation effect.





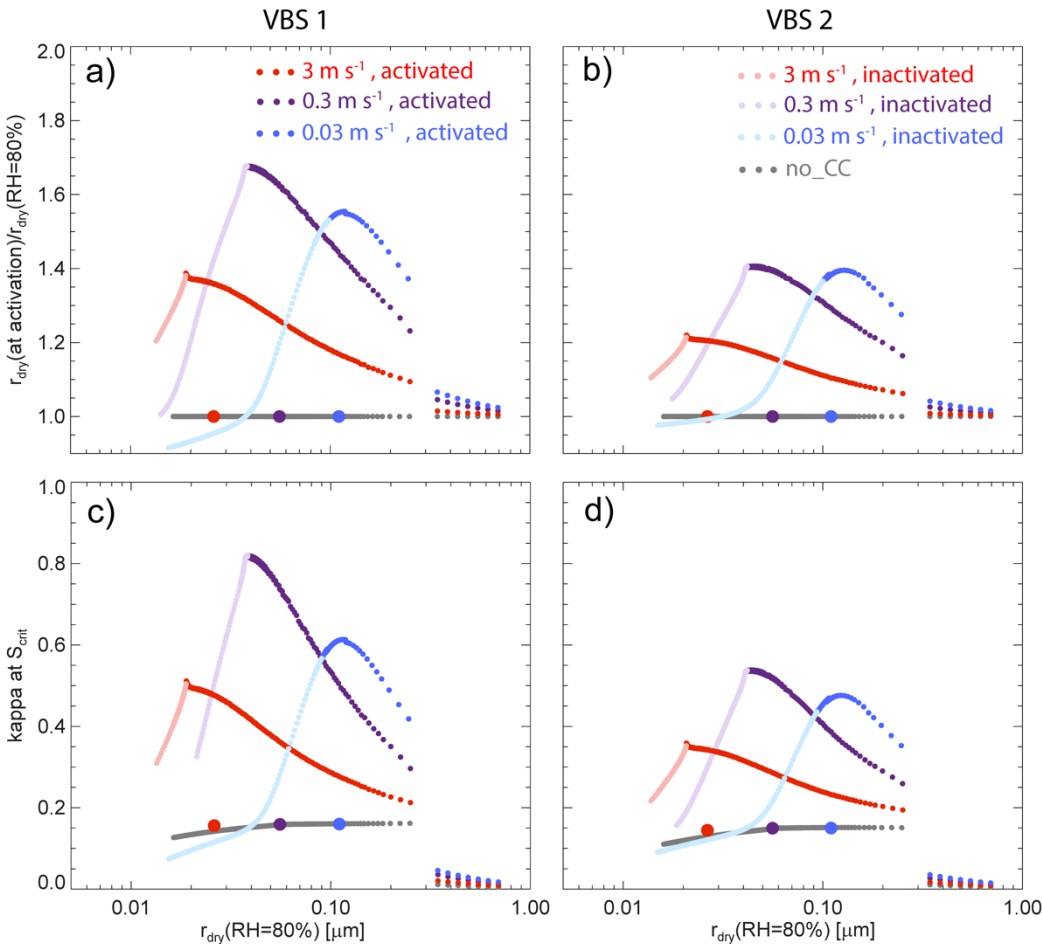

**Figure 8.** Dependence of relative dry radius (a, b) and equivalent $\kappa$ (c, d) at the activation point on initial particle size $r_{dry}$(RH = 80 %) for VBS 1 (a, c) and VBS 2 (b, d) for combined inorganic and organic co-condensation (coloured lines, co-condensation activated). The light portions of the lines represent non-activated particles evaluated at the time the last aerosol particle has activated. The grey line represents the control case without co-condensation for the updraft of 0.3 m s⁻¹. Coloured dots on the grey line mark the smallest activated dry radii of the control cases for the different updrafts.

To further explain these relationships, we plot the time evolution of key parameters for VBS 1 in Fig. S3 (VBS 2 in Fig. S4), the same way as in Fig. 6 for the three updraft velocities of 0.03, 0.3, and 3 m s⁻¹. This figure shows that with increasing updraft velocity the parcel uplifts faster and leaves less time for co-condensation. For VBS 1, the maximum water supersaturation is 0.07 %, 0.23 %, and 0.79 % for updrafts of 0.03, 0.3, and 3 m s⁻¹ (Fig. S3a), respectively, for which the critical radii at initial 80% RH are 104, 41, and 21 nm (Fig. 8a) marking the smallest radii of particles that activate. For updrafts of 0.03, 0.3, and 3



m s$^{-1}$, co-condensation results in a relative dry volume increase by a factor of 2.40, 2.13, and 1.35 at the activation point,
respectively, implying a lower co-condensed mass to enhance CDNC in the high updraft case. Although the updrafts of 0.03
and 0.3 m s$^{-1}$ exhibit a similar increase in dry mass, there is a clear difference in CDNC enhancement, because for the low
updraft of 0.03 m s$^{-1}$ most mass co-condenses on the larger particles, which activate anyway, while for larger updrafts, the
uptake shifts to smaller particles that would not activate without co-condensation. As the co-condensing mass at 3 m s$^{-1}$
partitions mostly to the small particles, the resulting CDNC enhancement is not much smaller than for 0.03 m s$^{-1}$ updraft,
despite the large difference in total co-condensing mass. Note that for low updrafts, the smallest activated particles deactivate
again while the supersaturation levels off to the value where its generation by the constant updraft is balanced by the water
vapor consumption through cloud droplet growth (see Figs. 6, S3 and S4). Overall, this analysis has shown that updraft
velocity, condensed and co-condensable mass, and aerosol size distribution are the key factors controlling the CDNC
enhancement due to co-condensation.

**3.3 Non-ideality of organic species**

Figure 9 shows the influence of the assumption of ideal mixing, which has been applied up to now as standard case in studies
simulating co-condensation (Topping et al., 2013; Heikkinen et al., 2024). To this end, we compare the CDNC change at
various updraft velocities for VBS 1 assuming ideality on one hand (panel a) and accounting for solution non-ideality (panel
b) on the other hand. Assuming ideality, activity coefficients are 1 for all species, while for non-ideality, the activity
coefficients of most organic species in a dilute aqueous solution are above 1 with values depending on their hydrophobic nature
as shown in Fig. 3. As a result, the assumption of solution-ideality overestimates co-condensation of organic species, while
they are in reality driven less to the condensed phase, resulting in a lower co-condensation effect on CDNC (green and blue
lines in Fig. 9). For inorganic species, the opposite is the case as their activity coefficients in dilute aqueous solution are below
1 and therefore, they are more driven to the condensed phase, resulting in a slightly higher co-condensation effect (orange lines
440   in Fig. 9). As overestimation of co-condensation for organics dominates the underestimation for inorganic species, an overall
overestimation of co-condensation results when non-ideality is ignored (blue lines). This shows that taking non-ideality into
account to estimate the co-condensation effect is important (maximum CDNC changes are 131% and 52% for ideal and non-
ideal cases, respectively), especially due to the overestimation of co-condensation for LO-OOA as these are rather
hydrophobic.



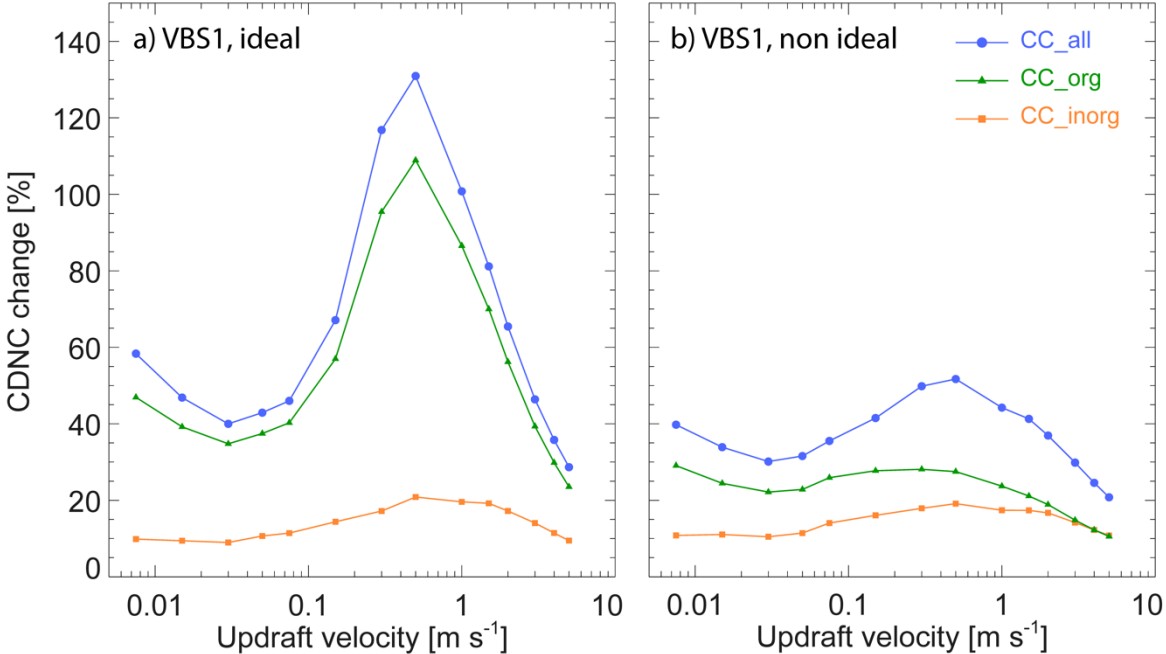

**Figure 9.** Change of CDNC as a function of updraft velocity for VBS 1, assuming solution ideality (panel a) and non-ideality (panel b) of inorganic-organic aqueous mixtures.

## 4 Conclusions

The CDNC change due to the co-condensation effect is sensitive to the initialisation conditions, especially the ambient surface temperature, aerosol size distribution, assumed volatility basis set of organics, non-ideality of inorganic-organic aqueous mixtures, and updraft velocities. We found that evaporation of organic mass caused by temperature change, heating and drying during sampling in the Q-ACSM instrument can lead to noticeable mass loss that should be considered in future studies for mass quantification and to estimate the co-condensation effect. It is worth noting that the evaporation loss is sensitive to various factors and can be decreased by using higher flow rates as e.g. in Heikkinen et al. (2020). The assumptions of the volatility basis set of organics are critical and highly uncertain due to technical limitation. Further developments in sensitivity calibration and quantification with FIGAERO-CIMS could help constraining the volatility basis set with $\log(C^*) \geq 4$. Non-ideality of inorganic-organic mixtures should be considered in co-condensation estimates as the ideality assumption could result in a significant overestimation of the CDNC enhancement, e.g., maximum CDNC change are 131% and 52% for ideal and non-ideal cases, respectively. The co-condensation effect is highly nonlinear and can vary significantly with environmental conditions. Further exploration with cloud parcel models in different environments, such as urban, rural, and marine regions,



are the first step to increase the understanding of the contributing factors and offers a basis to map the co-condensation effect over a regional and even global scale.

*Code and data availability.* The code can be obtained from Dr Beiping Luo (beiping.luo@env.ethz.ch).


*Author contributions.* C.M., Y.W., J.K., and B.L. designed the study and methodology. Y.W. procured the input data, the visualisations, and wrote the original draft. J.K. developed the VBS including non-ideality and AIOMFAC calculations. B.L. performed the cloud parcel model simulations and visualisations. G.C., L.H., and M.E. provided measurement data for parcel model initialisation. All authors contributed to writing (review & editing), and all authors have approved the final version of

the paper.

*Competing interests.* The authors declare that they have no conflict of interest.

*Acknowledgements.* We thank Ulrike Lohmann (ETH Zurich), Thomas Peter (ETH Zurich), Ying Chen (University of

Birmingham), and Mikael Ehn (University of Helsinki) for useful discussions and feedback on the manuscript. We acknowledge SMEAR II, INAR RI and ACTRIS infrastructure and personnel for the meteorological and aerosol size distribution data accessed from SmartSMEAR (https://smear.avaa.csc.fi/) and Krista Luoma (Finnish Meteorological Institute and University of Helsinki) for the Hyytiälä BC data (accessed through Chen et al., 2022).

*Financial support.* Y.W. is grateful for support from P. Sarasin and the ETH Zurich Foundation (ETH Fellowship project: 2021-HS-332) and the start-up fund for lectureship from University of Edinburgh. J. K. and B. L. have been funded by the Swiss National Science Foundation (SNF; grant no. 200021L_197149) as part of the ORACLE project.



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
