# Peer review of "Cloud droplet number enhancement from co-condensing NH3, HNO3, and organic vapours: boreal case study"

_EGUsphere, 2025_

## Author Comment (AC1)

**Authors' response to reviewers' comments for "Cloud droplet number enhancement from co-condensing NH$_3$, HNO$_3$, and organic vapours: sensitivity study"** by Yu Wang, Beiping Luo, Judith Kleinheins, Gang I. Chen, Liine Heikkinen, and Claudia Marcolli

We thank the reviewer for the suggestions that have significantly improved our manuscript. Below, we provide our responses and summarise the changes that we have made with line numbers referring to the uploaded document with tracked changes. Other minor revisions were also made to improve the manuscript.

This study investigates the impact of co-condensation of semi-volatile organic and inorganic compounds on cloud droplet formation in a boreal forest setting. Using a cloud parcel model that incorporates non-ideal mixing behavior, the authors demonstrate that combined co-condensation can enhance cloud droplet number concentration (CDNC) by up to 52% under realistic atmospheric conditions. Notably, the combined effect exceeds the sum of individual contributions, underscoring the critical role of organic volatility distributions and environmental parameters in cloud activation. While the study presents novel and compelling findings, the scope of conditions explored remains somewhat limited. Accounting for the non-ideality of organic compounds adds valuable realism to the simulations. Given the magnitude of the observed effect, it should be detectable through closure studies, and it is hoped that modelling efforts like this will inspire and guide such observational campaigns. I recommend the manuscript for acceptance, provided the detailed comments below are adequately addressed.

We thank the reviewer for their interest in our study.

Title: The title may be slightly overstated. Since only one size distribution is used and temperature is not varied, the sensitivity analysis is limited, making it difficult to draw firm conclusions about the role of different co-condensing gases. This is more like a case study.

The title was also a concern of Reviewer 2. We therefore revise it to:

Cloud droplet number enhancement from co-condensing NH$_3$, HNO$_3$, and organic vapours: boreal case study

Line 20: Replace "VBS" with "VBS distribution" for clarity. Replaced (line 21)

Line 35: Köhler (1936) is cited twice in the same sentence—consider revising to avoid redundancy. Second reference deleted (line 36)

Line 41: Note that Köhler theory has also been modified to include condensable gases. See: Laaksonen, A., Korhonen, P., Kulmala, M., and Charlson, R. J. (1998): Modification of the Köhler equation to include soluble trace gases and slightly soluble substances, J. Atmos. Sci., 55, 853–862.

We thank the reviewer for pointing out this study. We cite it in the revised paper (line 42) by revising: "...enhancing hygroscopic growth, modifying the Köhler curve (Laaksonen et al., 1998), and influencing cloud formation, ..."

Line 83: Why are only organics considered? Semivolatile inorganics are also likely to evaporate and should be addressed in a similar manner.

We also included co-condensation of inorganics as becomes clear later in the manuscript. To make this clear already in the introduction, we modify the introduction on lines 83–85: "Here, we set up two different VBS by integrating information from different experimental and modelling studies to investigate the sensitivity of the combined co-condensation effect of organic and inorganic species on VBS distribution."

Line 136: Typo in "Hyyitälä" – should be corrected to "Hyytiälä". Corrected (line 139)

Line 137: Is there a reference to support this assumption? A significant fraction of the semivolatile mass originates from this bin, so the assumption has a notable impact on the manuscript's conclusions.

We agree with the reviewer that this assumption has a notable impact. Assuming that the mass in volatility bin $\log(C^*) = 4$ is a result of secondary organic aerosol (SOA) formation, a high mass in this bin is justified, because SOA formation starts from volatile compounds, which are oxidized stepwise in a cascade of reactions to first intermediate, then semi-volatile and finally low volatility products. A suitable reference is Stolzenburg et al. (2022), which we cite now in the revised manuscript. We add on line 141:

"Note that this extrapolation leads to the highest mass fraction in volatility bin with $\log(C^*) = 4$. This high mass fraction is justified considering the cascade process of secondary organic aerosol formation starting from volatile compounds (Stolzenburg et al., 2022)."

Line 272: Does this assumption influence the results? Although the number of larger particles is small, their volume is substantial, which could affect the partitioning of semivolatile compounds.

We introduced the mineral dust to reach consistency between the aerosol mass measured by the Q-ACSM and the aerosol volume derived from the size distribution measured by the DMPS. As mineral dust typically belongs to the coarse aerosol fraction, the assumption that it forms the tail of the particle size distribution is justified. We assumed an organic coating on the mineral dust to allow absorptive partitioning. Following reviewer 2, we performed an additional run where we distributed the mineral dust more evenly among the particles. The figure below compares the change of CDNC as a function of cooling rate for the assumption that all particles with radius $r > 0.2\ \mu m$ possess a coating of 50 nm thickness with the assumption in the paper, namely that particles with $r > 0.25\ \mu m$ possess a coating of 10 nm thickness. It can be seen that the difference in the mineral dust distribution hardly affects the CDNC.

[Figure]

Lines 275–280: Some semivolatiles are already partitioned into particles at 80% RH. It would be helpful to show the aerosol size distribution before and after initial equilibration—does it still match observations?

We did this simulation and show it in the figure below. As the aerosol is dried before the measurement of the size distribution, we tailored our initialization to match the measurement at RH = 20% (black dashed line for measurements and solid lines for simulation). The size distribution at 80% was obtained by first equilibrating the aerosol with water vapor so that the vapor pressure including Kelvin effect above the particles corresponds to 80% RH (blue line). In a second step, both co-condensation of water vapor and semi-volatile species was performed at 80% (red line). As expected, the size distribution shifts to larger diameters mainly due to the water uptake.

[Figure]

Also, the choice of updraft velocity affects semivolatile partitioning. Why is a higher updraft used at RH below 98%? While the number of simulations is limited and computational cost is a factor, the model is a box model with parameterized thermodynamics and thus full simulations should be doable.

Repeating some simulations without using a higher updraft up to 98% RH, we realized that this indeed influenced the resulting CDNC. We therefore repeated all simulations starting from 80% RH with the correct updraft and updated the concerned figures and the numbers given in the text accordingly.

Repeating the calculations, we realized in addition that we made an error in our calculation of the ideal case. Namely, we used a too short equilibration time (only 2 min instead of the 30 min that we used for all other simulations). Because of this, there was too little partitioning to the condensed phase at 80% RH, and the gas-phase concentrations of organics still available for condensation was too high. Therefore, the CDNC increase resulting from co-condensation was overpredicted in Fig. 9 for the ideal case. Correcting this, the difference between ideal and non-ideal simulations became significantly smaller. We replaced Fig. 9 and changed the text accordingly.

Line 332: "Dry radius" seems incorrect here—this appears to refer to the wet radius, as water cannot be part of the dry radius definition.

We defined the use of "dry radius" and "dry volume" at the beginning of this paragraph (lines 334–336) by stating: "The co-condensation of organic and inorganic compounds increases the solute mass of the droplets, which can be expressed as a relative increase in dry volume and dry radius of the droplets compared to the initial state at ground level (time = 0 s, RH = 80%)." So, the dry mass at high RH is the sum of initial and co-condensed mass,

yet, without water. On line 332 (revised manuscript line 340), we use "dry radius" in this sense.

Figure 6: Is the observed change related to particle size before or after initial equilibration?

The figure starts after initial equilibration at 80% RH.

Line 362: Do simulations without co-condensation include the same initial equilibration step and after that the condensation is not allowed?

Yes. The initialization is always the same. We added this information to the revised manuscript by revising the sentence on lines 281–282 to:

"For all simulations including the control runs, the aerosol is equilibrated for 30 min at ambient conditions (80% RH) with the total gas phase given in Table 1 (inorganic species) and estimated in Sect. 2.4.1 (organic species)."

Line 397: "Interestingly…"—is this truly unexpected? The smallest (or nearly smallest) activating particles are also the most diluted, so this result seems intuitive.

Thank you for raising this question. We indeed think that this almost coincidence is interesting. Yet, the growth in $r_{dry}$ and the increase in kappa is not due to dilution in the strict sense as the dry particle growth is due to uptake of semi-volatile species. Moreover, the dilution is the same for particles of all sizes as it occurs in equilibrium with the gas-phase water vapor.  We think that it might be obvious that the smallest activating particles are the most grown among the activating particles, we do not find it obvious that they are also the most grown ones compared with the non-activating particles, if ones considers that the evaluation occurs at the point of activation for the activating particles and at the point when the last one has activated for the non-activating. So, particle growth due to co-condensation is an interplay between the Kelvin effect, which limits growth of small particles and diffusion limitation, which limits the growth of larger ones. We do not find it obvious that the maximum in growth due to this interplay almost coincides with the delimitation between activating/non-activating particles.

Line 428: "Key factors"—in practice, the sensitivity analysis in this paper only considers updraft velocity and two VBS distributions. Aerosol size distribution is not varied. What about temperature?

We did not vary temperature. Yet, we analysed the role of size distribution in the discussion of the figures by relating the increase in CDNC to the form of the size distribution. To make clearer that our conclusion about the relevance of the size distribution is a result of the analysis of the figures, we revise the sentence to (lines 444–446):

"Overall, the analysis of Figs. 6–8 has shown that updraft velocity, condensed and co-condensable mass, and aerosol size distribution are the key factors controlling the CDNC enhancement due to co-condensation."

**References**

Stolzenburg, D., Wang, M., Schervish, M., and Donahue, N. M.: Tutorial: Dynamic organic growth modeling with a volatility basis set, J. Aerosol Sci., 166, 106063, https://doi.org/10.1016/j.jaerosci.2022.106063, 2022.

---

## Author Comment (AC2)

**Authors' response to reviewers' comments for "Cloud droplet number enhancement from co-condensing NH₃, HNO₃, and organic vapours: sensitivity study"** by Yu Wang, Beiping Luo, Judith Kleinheins, Gang I. Chen, Liine Heikkinen, and Claudia Marcolli

We thank the reviewer for the suggestions that have significantly improved our manuscript. Below, we provide our responses and summarise the changes that we have made with line numbers referring to the uploaded document with tracked changes. Other minor revisions were also made to improve the manuscript.

**Summary**

This paper uses a non-ideal cloud parcel model (with AIOMFAC/Pitzer activity treatment) to quantify CDNC changes from co-condensation of semi-volatile organics and inorganics for a Hyytiälä boreal case. The core findings are: (i) combined co-condensation raises CDNC by up to ~52% in the non-ideal case (vs 131% if ideal mixing is assumed), and (ii) the combined effect exceeds the sum of organics-only and inorganics-only contributions, with the largest boosts at intermediate updrafts. The setup uses a single observed size distribution and fixed ambient state (≈8 °C, 80% RH).

We thank the reviewer for their interest in our study.

**Assessment**: interesting, well-motivated case study; methods are appropriate; conclusions are supportable with some clarifications. I recommend accept after minor revision.

Strengths

- Clear demonstration that non-ideality matters and that assuming ideality overestimates the organic contribution to co-condensation and thus CDNC.

- Sensible separation of organic vs inorganic roles and a transparent VBS framing, including the importance of higher-volatility bins (log $C^* \approx 4$) near activation.

- Mechanistic analysis across updrafts showing a 21–52% CDNC increase for combined organics+inorganics and a non-linear dependence on w.

**Essential clarifications**

1. **Title scope**
   Current title reads more general than the experiments (single size distribution; fixed T).
   **Action:** tone down to "boreal case study" in title or abstract.
   Example: "Synergistic organic–inorganic co-condensation enhances CDNC in a boreal case study with non-ideal mixing."

   The title was also a concern of Reviewer 2. We therefore revise it to:
   "Cloud droplet number enhancement from co-condensing NH₃, HNO₃, and organic vapours: boreal case study"

2. **Parcel-model upper bound / entrainment context**
   Field CDNC is often lower than parcel-model CDNC because entrainment, w-variability,

and turbulent quenching reduce Smax and can deactivate marginal droplets; semi-volatiles taken up near activation can re-evaporate upon mixing.

**Action:** add 2–3 sentences in Discussion stating that reported CDNC enhancements are an upper bound for in-cloud conditions and that entrainment could buffer these effects.

We thank the reviewer for pointing out this aspect. We add the following sentence to the conclusions on lines 351–353:

"Note that our simulations describe cloud droplet activation at cloud base and do not include turbulence and entrainment-mixing, which can also lead to deactivation of cloud droplets thus affecting CDNC and droplet size distributions (Morales et al., 2011; Yang et al., 2018; Oh et al., 2023)."

**Initialisation & computational shortcut**

The manuscript uses 1.2 m s⁻¹ below 98% RH then switches to the target w, claiming negligible impact because most uptake occurs above 98% RH. This is fine, but please make this explicit in explaining the control case too (if that's what was done)

**Action:** state clearly that control (no co-condensation) runs use the same 80%→equilibration step (I know it is simpler to equilibrate the aerosol when there are no other co-condensing vapours) and the same pre-98% RH shortcut, and add one sentence reporting that a no-shortcut check (e.g. for at least one of the co-condensing cases) produced CDNC within X%.

We improved the description of the initialisation procedure by revising starting from line 245:

"The concentrations of inorganic salts and gaseous ammonia and nitric acid are taken from Table 1. To match the total mass concentration according to the size distribution, the aerosol composition was complemented with Na+, mineral dust and black carbon as described in Sect. 2.4.2. This composition initialisation was applied to all simulations, including the control runs."

**Non-ideality framing**

You nicely document that ideality inflates CDNC changes (131% vs 52%). Consider reporting the range too, rather than just the maximum for ideal vs non-ideal CDNC (and Smax) with error bars for the range (e.g. the data in figure 9 suggests 25 to 131% (ideal) vs 25 to 52% (non-ideal)) for quick reader digestion.

Thanks for this suggestion. We now refer to the range by revising (line 451):

"...(CDNC changes are 14–53% and 20–44% for ideal and non-ideal cases, respectively)..."

Note that the revised numbers are considerably smaller than the ones reported in the manuscript. This is because we realized an error in our calculation for the ideal case during the revisions. Namely, we had used a too short equilibration time (only 2 min instead of 30 min). Because of this, there was too little partitioning to the condensed phase at 80% RH.

Due to the high gas-phase concentrations of organics still available for condensation, the CDNC increase resulting from co-condensation was overpredicted in Fig. 9. Correcting this, the difference between ideal and non-ideal simulations became significantly smaller.

**Large-particle/composition completion**

You add mineral dust (10% v/v; r > 250 nm with a 10 nm coating) and BC (3.6% v/v) to close mass/volume. Briefly say whether toggling this coarse tail alters partitioning/CDNC (expected: small).

**Action:** add a clause like "Removing/halving the coarse tail changed CDNC by <X%, indicating little influence on the results."

We introduced the mineral dust to reach consistency between the aerosol mass measured by the Q-ACSM and the aerosol volume derived from the size distribution measured by the DMPS. As mineral dust typically belongs to the coarse aerosol fraction, the assumption that it forms the tail of the particle size distribution is justified. We assumed an organic coating on the mineral dust to allow absorptive partitioning. Following reviewer 1, we performed an additional run where we distributed the mineral dust more evenly among the particles. The figure below compares the change of CDNC as a function of cooling rate for the assumption that all particles with radius r > 0.2 μm possess a coating of 50 nm thickness with the assumption in the paper, namely that particles with r > 0.25 μm possess a coating of 10 nm thickness. It can be seen that the difference in the mineral dust distribution hardly affects the CDNC.

[Figure]

**Detectability statement**

The abstract suggests the magnitude should be detectable in closure studies. Please outline a practical closure strategy for observations.

We add the following statement to the conclusion.

"Note that our simulations describe cloud droplet activation at cloud base and do not include turbulence and entrainment-mixing, which can lead to deactivation of cloud droplets thus affecting CDNC and droplet size distributions (Morales et al., 2011; Yang et al., 2018; Oh et al., 2023). This will also render the observation of this effect in field measurements more difficult.  To detect it in closure studies, the measurements should be performed close to cloud base, while cloud edges where entrainment is most likely to affect CDNC (Freud et al., 2011) should be avoided."

**Specific:**

- Line 35: avoid citing Köhler (1936) twice in the same sentence. We removed one reference.

- Line 136: fix "Hyyitälä" → Hyytiälä. fixed

- Line 137: extrapolation. What justification is there for this extrapolation? And how was the extrapolation done? What method? straight line? Last two bins, or some kind of mass closure? I know you say bins 0-3 but more detail needed.

  We agree with the reviewer that this assumption has a notable impact. We think that a high mass in volatility bin as a result of secondary organic aerosol formation is justified as this process starts from volatile products, which are oxidized stepwise in a cascade of reactions to first intermediate, then semi-volatile to finally low volatility compounds. A suitable reference for this notion is Stolzenburg et al. (2022), which we cite now in the revised manuscript. We add after "VBS that we derived": "Note that this extrapolation leads to the highest mass fraction in the volatility bin with log(C*) = 4. This high mass fraction is justified considering the cascade process of secondary organic aerosol formation starting from volatile compounds (Stolzenburg et al., 2022)."

- Line 272: explicitly state whether assumptions about large particles/coatings influence partitioning and CDNC

  See the response to comment "Large-particle/composition completion".

- Lines 275–280: show (or state) that the post-equilibration size distribution still matches DMPS within uncertainty; justify the high-w shortcut with the quick check for one or two cases.

- We did this simulation and show it in the figure below. As the aerosol is dried before the measurement of the size distribution, we tailored our initialisation to match the measurements at RH = 20% (black dashed line for measurements and solid lines for simulation). The size distribution at 80% was obtained by first equilibrating the aerosol with water vapor so that the vapor pressure including Kelvin effect above the

particles corresponds to 80% RH (blue line). In a second step, both, co-condensation of water vapor and semi-volatile species was performed again at 80% (red line). As expected, the size distribution shifts to larger diameters mainly due to the water uptake.

[Figure]

- Line 428 ("Key factors"): list exactly what you varied (VBS, w, non-ideality) and note temperature/size-distribution variability as likely important but not explored here.

We did not come to this summary of key factors through direct variation of parameters, but through the analysis of Figs. 6-8. We make this clear by revising the sentence to (lines 444–446):

"Overall, the analysis of Figs. 6–8 has shown that updraft velocity, condensed and co-condensable mass, and aerosol size distribution are key factors controlling the CDNC enhancement due to co-condensation."
Moreover, we extend the conclusion starting from line ...:
"Overall, systematic variation of updraft velocity during cloud droplet activation allowed elucidating how the aerosol size distribution influences the enhancement of CDNC due to co-condensation. Simulations with two different VBS showed that the mass in the bin with log(C*) = 4 has a large influence on the magnitude of the co-condensation effect. Inclusion of non-ideality proved to be relevant for a realistic estimate of the co-condensation effect especially for the less oxidized organics. Studies in other regions with different aerosol size distribution and composition are required to establish more comprehensively the role co-condensation plays in cloud droplet activation.
Note that our simulations describe cloud droplet activation at cloud base and do not include turbulence and entrainment-mixing, which can lead to deactivation of cloud droplets thus affecting CDNC and droplet size distributions (Morales et al., 2011;

Yang et al., 2018; Oh et al., 2023). This will also render the observation of the co-condensation effect in field measurements more difficult. To detect it in closure studies, measurements should be performed close to cloud base, while cloud edges where entrainment is most likely to affect CDNC (Freud et al., 2011) should be avoided."

References

Freud, E., Rosenfeld, D., and Kulkarni, J. R.: Resolving both entrainment-mixing and number of activated CCN in deep convective clouds, Atmos. Chem. Phys., 11, 12887–12900, https://doi.org/10.5194/acp-11-12887-2011, 2011.

Morales, R., Nenes, A., Jonsson, H., Flagan, R. C., and Seinfeld, J. H.: Evaluation of an entraining droplet activation parameterization using in situ cloud data, J. Geophys. Res., 116, D15205, https://doi.org/10.1029/2010JD015324, 2011.

Oh, D., Y. Noh, and F. Hoffmann, 2023: Paths from aerosol particles to activation and cloud droplets in shallow cumulus clouds: The roles of entrainment and supersaturation fluctuations. J. Geophys. Res. Atmos., 128, e2022JD038450, https://doi.org/10.1029/2022JD038450.

Stolzenburg, D., Wang, M., Schervish, M., and Donahue, N. M.: Tutorial: Dynamic organic growth modeling with a volatility basis set, J. Aerosol Sci., 166, 106063, https://doi.org/10.1016/j.jaerosci.2022.106063, 2022.

Yang, F., Kollias, P., Shaw, R. A., and Vogelmann, A. M.: Cloud droplet size distribution broadening during diffusional growth: ripening amplified by deactivation and reactivation, Atmos. Chem. Phys., 18, 7313–7328, https://doi.org/10.5194/acp-18- 7313-2018, 2018.

---

## Author Response (AR2)

**Second revision:**

**Authors' response to reviewers' comments for "Cloud droplet number enhancement from co-condensing NH₃, HNO₃, and organic vapours: boreal case study"** by Yu Wang, Beiping Luo, Judith Kleinheins, Gang I. Chen, Liine Heikkinen, and Claudia Marcolli

**Comments of Reviewer 1:**

My original comment "Why are only organics considered? Semivolatile inorganics are also likely to evaporate and should be addressed in a similar manner." may have been misunderstood. What I intended to refer to was the initialization of the gas phase in the simulations. Since there were no gas-phase observations of $HNO_3$ and $NH_3$ available during the campaign, I was wondering why values from previous years were used, rather than applying the same approach as was done for the organic species. Are the inlets in Makkonen et al (2014) for gas phase concentrations similar to the ones used in aerosol data in the present manuscript? I understand that the main focus of the manuscript is on the co-condensation of organics; however, the role of inorganics remains highly uncertain, and this uncertainty can also affect the inferred relative contributions.

Hyytiälä is still misspelled on line 139.

**Responses:**

This study was intended to explore factors that influence – for a realistic environmental situation – co-condensation of organics rather than to reproduce exactly the situation in Hyytiälä during the autumn of 2018. As such, it is between a case study and a sensitivity study. This was the main reason why we decided to use the values measured during 2010 instead of simulating the evaporation of $HNO_3$ and $NH_3$ during sampling. Moreover, the deduction of the total concentration (gas and condensed phase) of organic and inorganic species from the condensed-phase concentrations only would be very hard to bring to convergence as many interdependent species are involved, whose total concentration must be varied until the simulated condensed-phase concentrations match the measurements.

We have corrected the typo in Hyytiälä on line 139.